# Associations of Vitamin D Levels with Physical Fitness and Motor Performance; A Cross-Sectional Study in Youth Soccer Players from Southern Croatia

**DOI:** 10.3390/biology10080751

**Published:** 2021-08-05

**Authors:** Barbara Gilic, Jelena Kosor, David Jimenez-Pavon, Josko Markic, Zeljka Karin, Daniela Supe Domic, Damir Sekulic

**Affiliations:** 1Faculty of Kinesiology, University of Split, 21000 Split, Croatia; barbaragilic@gmail.com; 2Faculty of Kinesiology, University of Zagreb, 10000 Zagreb, Croatia; 3Department of Pediatrics, University Hospital of Split, 21000 Split, Croatia; jelena.kosor1@gmail.com (J.K.); jmarkic@mefst.hr (J.M.); 4MOVE-IT Research Group, Department of Physical Education, Faculty of Education Sciences, University of Cadiz, 11519 Cadiz, Spain; david.jimenez@uca.es; 5Biomedical Research and Innovation Institute of Cadiz (INiBICA) Research Unit, Puerta del Mar University Hospital University of Cadiz, 11009 Cadiz, Spain; 6CIBER of Frailty and Healthy Aging (CIBERFES), 28001 Madrid, Spain; 7School of Medicine, University of Split, 21000 Split, Croatia; 8Teaching Institute of Public Health of Split Dalmatian County, 21000 Split, Croatia; karinzeljka@gmail.com; 9Department of Medical Laboratory Diagnostics, University Hospital of Split, 21000 Split, Croatia; daniela.supedomic@gmail.com; 10Department of Health Studies, University of Split, 21000 Split, Croatia

**Keywords:** 25(OH)D, physiology of performances, puberty, pre-planned agility, non-planned agility

## Abstract

**Simple Summary:**

Vitamin D is a fat-soluble prohormone crucial for bone mineralization, muscle contractility, and neurological conductivity. It is theorized that Vitamin D plays an important role in sport performances, especially in young athletes. In this study we examined the associations of levels of 25-hydroxyvitamin D (25(OH)D) with physical fitness and motor-performance achievements in youth soccer players from Southern Croatia. Participants were tested on physical fitness, motor performance and vitamin D at the end of the winter period, when levels of vitamin D are known to be lowest due to low exposure to sunlight. Results showed that deficiency of 25(OH)D was widespread among youth soccer players living in Southern Croatia. Low 25(OH)D levels were associated with lower results in fitness tests (i.e., tests of energetic capacities), but there was no correlation between 25(OH)D levels and the results in motor performance tests (i.e., skill tests). Our results support the theory of the association between vitamin D and energetic capacities of athletes, but there is no evidence on association between vitamin D and skill-based capacities.

**Abstract:**

Vitamin D level is known to be a factor potentially influencing physical fitness, but few studies have examined this phenomenon among youth athletes. We aimed to evaluate the associations of vitamin D levels (as measured by 25-hydroxyvitamin D concentrations—25(OH)D) with various physical fitness and motor performance tests in youth football (soccer) players. This cross-sectional study included a total of 52 youth soccer players (15.98 ± 2.26 years old) from Southern Croatia. The participants were evaluated at the end of the winter period and data were collected of anthropometric measures (body mass and body height), vitamin D status (25(OH)D levels), physical fitness tests (sprints of 10 and 20 m, 20 yards test, the countermovement jump, the reactive strength index (RSI)) and motor performance tests (the soccer-specific CODS, the soccer-specific agility, and static balance). Among the studied players, 54% had 25(OH)D insufficiency/deficiency, showing a lack of 25(OH)D is widespread even in youth athletes living at a southern latitude. The 25(OH)D level was correlated with sprint 20 m, 20 yards tests, and RSI, showing a greater role of 25(OH)D in physical fitness tests where energetic capacity is essential than in sport-related motor performance tests where skills are crucial. Our results support the idea that vitamin D can play a determinant role in physical fitness tests with a clear physiological component, but is not crucial in motor performance tests related to specific sports where skills are a key component. Future studies should investigate the effects of vitamin D supplementation on the performance in physical fitness and motor performance tests among youth athletes.

## 1. Introduction

Vitamin D is a fat-soluble prohormone with the function of maintaining the concentrations of calcium and phosphate within the physiological ranges, which is crucial for bone mineralization, muscle contractility, and neurological conductivity [1]. A total of 80–90% of vitamin D is synthesized during exposure to sunlight ultraviolet B radiation, whereas 10–20% is obtained from food [2]. Many factors influence vitamin D status including age, season, latitude, nutrition, physical activity levels, and body-fat percentage [3]. The circulating form of vitamin D, 25(OH)D, is often used for determining its status in the human body, and is considered the most accurate indicator of cutaneous synthesis and nutritional intake [4,5].

During the last few decades, interest has increased in vitamin D research because it was discovered that almost every body tissue has a vitamin D receptor, meaning that this can directly and indirectly influence their functions [6]. Specifically, vitamin D affects the regulation of the differentiation, proliferation, and growth of cells; hormone production; and immune, nervous, and muscle systems [7]. Regarding these functions, vitamin D is considered to play a role in optimal sports performance since it is involved in muscle physiology as muscles express a high number of vitamin D receptors, affects the transport of phosphate and calcium across muscle cell membranes, modulates phospholipid metabolism, and induces the expression of several myogenic transcription factors and myotubular sizes, which together affect the contractile filaments [8,9,10].

Simultaneously, evidence points to suboptimal vitamin D status in the general population, including athletes, children, and adolescents [11,12,13]. Collectively, studies frequently evidence inadequate levels of 25(OH)D among athletes worldwide [13]. Athletes are more susceptible to being vitamin D deficient/insufficient compared with the general population, probably because of their increased enzymatic activity following exercise [14]. The problem is additionally accentuated in youth athletes, since adolescents have an increased risk of malnutrition due to an increased need for energy and nutrients required for proper growth and development [15,16].

Soccer (football) is one of the most popular sports, characterized by a combination of low- and high-intensity activities, alternating short periods of high intensity activity with long periods of low intensity [17,18]. Although the aerobic metabolic component prevails in the form of low- and medium-intensity running, its high-intensity (anaerobic) activities, such as sprints, jumps, stopping, changes of direction, and striking, are key determinants of game outcomes [19]. As these activities are crucial for soccer, studies have already investigated the association between the status of vitamin D and various performance capacities in this sport. A significant correlation was found between vitamin D levels and muscle functioning assessed with jumping, sprinting, leg press, and aerobic capacity tests in professional Greek soccer players [9]. Alimoradi et al., (2019) also recorded a positive correlation between higher vitamin D concentration and improvements in sprint and leg press tests in a group of Iranian soccer players supplemented with vitamin D in comparison with players who were not supplemented [20]. Conversely, Ksiazek et al., (2016) did not find an association between vitamin D levels and lower limb muscle strength in professional soccer players [21]. 

However, several studies conducted on younger soccer players have reported inconsistent results. Low vitamin D levels were associated with low muscle strength level, changes in direction, jumps, and sprints in 7- to 15-year-old children involved in soccer training [22]. A more recent study by Bezrati et al. (2020) recorded significant improvements in sprinting, change in direction, and running speed tests after vitamin D supplementation in youth soccer players aged 8–15 years who were vitamin deficit at the beginning of the research [23]. Bezuglov et al. (2019) also noted a weak association of 25(OH)D level with running speed and muscle strength among soccer players aged 13–18 years [24]. Moreover, after 8 weeks of vitamin D supplementation, Jastrzebska et al., (2016) and Skalaska et al. (2019) did not record any significant difference in sports performance between placebo and experimental groups of soccer players aged 16–18 years [25,26].

As even the smallest improvements in any aspect affecting sports performance can lead to improvements in sports results, it is important to further investigate the association between vitamin D and sports performance. However, as evidenced from our review of the literature, studies that examined the associations between vitamin D status and physical performance in youth soccer players reported inconsistent findings [22,24,25,26]. The authors of previous studies examined relatively narrow sets of physical performance (strength, jumping, and sprinting capacities), which are generally called tests of physical fitness. On the other hand, previous studies rarely observed any of the skill-based motor performance such as balance, sport-specific change of direction speed (soccer-CODS), and sport-specific reactive agility (soccer-AGIL), which are generally considered more important determinants of success in soccer [19,27]. Finally, as an important methodological issue it must be mentioned that previous studies involved players from different teams [22,24,25,26]. As a result, participants’ physical performance could vary due to differences in their training regimes and training methodologies, irrespective of vitamin D status. Therefore, the aims of this research were: (i) to determine the status of vitamin D and (ii) to evaluate the associations of 25(OH)D levels with physical fitness and motor performance in youth soccer players that were members of the same team living in Southern Croatia (Mediterranean region) at the end of the winter period.

## 2. Materials and Methods

### 2.1. Participants

The 52 participants in this cross-sectional research were youth soccer players (15.98 ± 2.26 years old). All players were members of the same soccer team located in Split, Croatia, residing at the latitude of 43° N Prior to the testing procedures, all players were informed of the purpose of the research and provided informed consent (for participants under 18 years of age, informed consent was signed by the parent or legal guardian). Inclusion criteria were active soccer training for at least three years and presence at 80% of the training sessions during the last month. The exclusion criteria were illness or injury that could have reduced the intensity of training during the last month and presence of pain in any part of the body during the testing. The testing was conducted during February 2020. The study was approved by the Ethical Board of the University of Split, Faculty of Kinesiology, and was conducted according to the guidelines in the newest version of the Declaration of Helsinki (approval No.: 2181-205-02-05-14-001).

### 2.2. Variables and Testing

The variables included in this study were anthropometric measures (body mass and body height), vitamin D status (25(OH)D levels), fitness tests, and motor performance tests.

Anthropometric variables were measured using standardized equipment by an experienced technician. Body height was measured in cm (accurate to 0.5 cm), and body mass (BM) accurate to 0.1 kg. Body mass index (BMI) was calculated using the following equation: BMI = BM (kg)/BH (m^2^).

The 25(OH)D levels were measured using the Elecsys vitamin D total assay (electro-chemiluminescence binding assay (ECLIA)), and with a Cobas e601 analyzer (Roche Diagnostics International Ltd., Rotkreuz, Switzerland), using a competitive electrochemiluminescence binding technique. The vitamin D total assay employs a vitamin D-binding protein as the capture protein to bind vitamin D3(25-OH) and vitamin D2(25-OH). The detection range of the test is 7.5–175 nmol/L 25(OH)D, and the sensitivity of the assay is 5 nmol 25(OH)D/L. The intraclass CV ranges from 2.2% (at 174 nmol 25(OH)D/L), to 6.7% (at 165 nmol 25(OH)D/L), with the 5.0 nmol/L, 7.5 nmol/L and 12.5 nmol/L for the limit of blank, limit of detection, and limit of quantification, respectively. The blood samples were taken from the athletes prior to the morning exercise session and were analyzed at the laboratory of the University Hospital of Split, Croatia. For the purpose of this study the 25(OH)D values of >75 nmol/L, 51–75 nmol/L, and <50 nmol/L were used for the classification of vitamin D sufficiency, insufficiency, and deficiency, respectively [28]. In further statistical analyses (please see later text for details) players were divided into two groups according to the 25(OH)D levels: inadequate levels of 25(OH)D (vitamin D deficiency/insufficiency, <75 nmol/L), and adequate levels of 25(OH)D (vitamin D sufficiency, >75 nmol/L).

Physical fitness tests included sprint over 10 and 20 m distance, generic test of change of direction speed (20 Yards test), countermovement jump, and reactive strength index.

The 10 (S10M) and 20 m (S20M) sprint tests were assessed by photoelectronic timing gates (Powertimer, Newtest, Oulu, Finland). The timing gates were placed 10 and 20 m from the starting line. Players were instructed to run as fast as possible from the start line along the 20 m distance. The time was recorded at 10 and 20 m. Players performed three testing trials and the best result was further analyzed.

The 20 yards test (20Y) was performed on a 10-yard field. One cone was placed in the center, the second 5 yards to the left, and the third 5 yards to the right from the center cone. Players were laterally standing 50 cm from the central cone where timing gate (Powertimer, Newtest, Oulu, Finland) was placed. Players had to rotate 90° and run 5 yards to the left, turn, sprint 10 yards to the right, turn and sprint 5 yards back to the central cone. Players performed three trials, and the best result was used for further analysis. For the countermovement jump test (CMJ), players started from the standing position with hands placed on their hips. They performed a downward movement to approximately 90° knee flexion, followed by a maximum upward movement. The height of the jump was measured using Optojump (Microgate, Bolzano, Italy). Participants performed three testing trials, and the best result was further analyzed.

The reactive strength index (RSI) was calculated from the jumped height and the ground contact time while performing the drop jump with both legs. The RSI was measured with the Optojump system (Microgate, Bolzano, Italy). The test was performed three times, and the best result was included in the analysis.

Motor performance tests included tests of soccer-specific change of direction speed test (soccer-CODS), soccer-specific agility test (soccer-AGIL), and balance test of overall stability index (OSI).

The soccer-CODS, and soccer-AGIL were tested with specifically designed tests. An infrared (IR) sensor was used as input for time triggering, and LEDs placed in the cones were set as outputs. The cones were shaping a Y pattern. The first cone was placed at the start line, and in that cone, IR sensor was placed. The other two cones were placed 4.5 m diagonal from the starting cone. For both tests, players had to run from the start line, switching the IR sensor, which triggered one of the lighting cones, and when the timing began. They had to run to the lit cone, kick the ball placed in front of that cone and run back to the start line when the time was recorded. For soccer-AGIL, the players did not know which cone would be lit, whereas, for the soccer-CODS, they knew the testing scenario in advance. For soccer-AGIL, the players performed five trials; for soccer-CODS, they performed two trials. The best results were used in the analysis for both soccer-CODS and soccer-AGIL [29] (Figure 1).

The OSI was measured using the Biodex Balance System. The resistance level was set to number 6, with trials lasting 20 s. Participants performed three trials and the best result was further included in the study [30].

### 2.3. Statistics

Variables were checked for normality of the distributions by the Kolmogorov–Smirnov test, and descriptive statistics included means and standard deviations. The test-retest reliability of the conditioning capacities was previously studied and reported in detail [29] and, therefore, in this study, all tests were checked for intratesting reliability by calculation of the intraclass coefficient (ICC) and coefficients of variation (CV).

To define the differences between the groups on the basis of vitamin D levels (vitamin D deficiency/insufficiency vs. vitamin D sufficiency), Student’s *t*-test for independent samples was applied and further analyzed using the magnitude-based Cohen’s effect size (ES) statistic with modified qualitative descriptors (ES ranges: <0.02 = trivial; 0.2–0.6 = small; >0.6–1.2 = moderate; >1.2–2.0 = large; and >2.0 very large differences) [31].

To identify the associations between 25(OH)D levels (measured values in nmol/L), and physical fitness and motor performance, Pearson’s product moment correlation coefficients were calculated. These analyses were undertaken in two phases. In the first phase we calculated simple univariate correlations and regression coefficients between all pairs of variables. In the second phase, the significant correlations (regression coefficients) between 25(OH)D and physical fitness/motor performances that were evidenced in the first phase were additionally statistically controlled. For this purpose we included age and BMI as covariates in the correlation analysis, and statistically controlled the confounding effect of age in calculated correlations between vitamin D and physical fitness/motor performance variables. This was done since analyses showed significant correlation between age and vitamin D levels in the first phase, while there was a possibility of the confounding effects of age on association between 25(OH)D and fitness/performance. Pearson’s product moment correlation coefficients (Pearson’s R) of 0.1–0.29; 0.3–0.49; 0.5–1.0 (positive and negative values) represented low, moderate, and large correlation, respectively [32].

The type I error rate of 5% (*p* < 0.05) was set a priori and was considered statistically significant. StatSoft Statistica ver. 13.0 (Tulsa, OK, USA) was used for all analyses.

## 3. Results

Results of descriptive statistics for study variables and reliability parameters for physical fitness and motor performance tests are presented in Table 1. The intratesting-reliability of the applied tests of conditioning capacities ranged from appropriate values for OSI (ICC = 0.76, CV = 11%), to high reliability for S10 and S20M (ICC = 0.90 and 0.94, CV = 3% and 4%, respectively).

Figure 2 presents the distribution of vitamin D sufficiency, insufficiency, and deficiency in youth players from southern Croatia, at the end of the winter period. In brief, 46.4% of players had sufficient levels of vitamin D (20(OH)D > 75 nmol/L), the vitamin D insufficiency (25(OH)D levels of 50–75 nmol/L) was evidenced in 44% of players, while 9.6% of studies players had vitamin D deficiency (25(OH)D < 50 nmol/L). The mean value for the 25(OH)D level was 79.03 nmol/L (standard deviation of 25.32), while median value was 73.55 nmol/L.

Table 2 presents differences between groups of players on the basis of vitamin D status. Significant differences between groups were found in age (*t*-test = 2.2, *p* = 0.03), participants with sufficient vitamin D levels were older. Also, significant differences were found for S20M *(t*-test = 2.45, *p* = 0.02), and 20Y (*t*-test = 2.16, *p* = 0.04). In both cases, better results were achieved by participants who had sufficient vitamin D levels.

Figure 3 presents ES differences between groups on studied variables according to their vitamin D status (sufficiency vs. insufficiency/deficiency). Moderate ES between groups based on 25(OH)D levels were evidenced for age (d = 0.65, 95%CI: 0.1–1.2), S20M (d = 0.68, 95%CI: 0.12–1.24), 20Y (d = 0.62, 95%CI: 0.06–1.17), while small ES were evidenced for all other variables.

Table 3 presents associations between study variables. Pearson’s correlations between studied variables showed significant positive associations among most of the conditioning capacities. The negative correlation coefficients in some cases are results of opposite metrics of the variables (i.e., better achievement in sprinting is noted with a numerically lower result, while CMJ is noted with a numerically higher result), and practically highlight positive correlations between capacities. On the other hand, correlations between vitamin D levels and conditioning capacities reached statistical significance (*p* < 0.05) for S20M (Pearson’s R = 0.39, low positive correlation), 20Y (Pearson’s R = 0.31, low positive correlation), CMJ (Pearson’s R = 0.27, low positive correlation), and RSI (Pearson’s R = 0.36, low positive correlation). Since vitamin D status was significantly (*p* < 0.05) correlated with participants’ age (Pearson’s R = 0.33, low positive correlation), the significant correlations between fitness variables (S20M, 20Y, CMJ, and RSI), and vitamin D status were additionally controlled for age as a covariate. Finally, significant (*p* < 0.05) partial correlations were confirmed for associations between vitamin D and (i) S20M (Pearson’s R = −0.30, low negative correlation), (ii) 20Y (Pearson’s R = −0.31, low negative correlation), and (iii) RSI (Pearson’s R = 0.32, low positive correlation).

## 4. Discussion

This study has several major findings. First, a deficiency of 25(OH)D was widespread among youth soccer players living in Southern Croatia. Second, low 25(OH)D levels were associated with lower results in the S20M, 20Y, and RSI (fitness tests). Last, there was no correlation between 25(OH)D levels and the results in motor performance tests.

### 4.1. Vitamin D Status in Youth Soccer Players

The mean value of serum 25(OH)D level of the studied players was 79.03 ± 25.32 nmol/L, with 54% of participants having low 25(OH)D levels (<75 nmol/L). These results are somewhat better than those with previously reported data in Croatia, neighboring countries, and countries with similar latitudes for samples of children and adolescents. For example, very recent report evidenced 38 ± 13 nmol/L in Italian, and 52 ± 14 nmol/L in Spanish children (42° N and 40° N, for Italian and Spanish samples, respectively) [33]. Furthermore, 72.3% Bosnian and Herzegovinian (B&H) adolescents (<18 years old) had low 25(OH)D levels (<75 nmol/L) [34], whereas 82.2% of Italian adolescents (10–21 years old) had hypovitaminosis D with a median serum 25(OH)D level of 50 nmol/L [35]. The somewhat better status of adolescents observed in our study can probably be attributed to the fact that our participants were athletes. Specifically, it is unlikely that adolescents with very low 25(OH)D levels will participate in systematic training (note that our participants had a minimum of 6 years of experience in soccer).

The results obtained are in line with several studies conducted on athletes worldwide. Specifically, a review study noted that 56% of athletes involved in different sports had inadequate 25(OH)D levels [13]. Similar results were noted for youth soccer players. A recent Russian study reported low 25(OH)D levels in 42.8% of youth soccer players [36] and 61.1% of youth Polish soccer players were found to have 25(OH)D concentration <50 nmol/L [26]. In addition to the general reasons for the lack of vitamin D (i.e., lack of vitamin D in regular nutrition, low bioavailability of the vitamin D) [15], another possible reason for low 25(OH)D levels could be the period or season in which the players were tested in our study (during February, which is the end of the winter). In brief, the skin synthesis of vitamin D is lower from October to March in regions far from the equator (i.e., above 35° north or south) [37]. As Croatia is located at 42°–46° N, the accumulation of vitamin D during the winter season is inadequate. In agreement with our findings, Morton et al. evidenced a significant decrease in vitamin D levels in professional English soccer players between the summer and winter seasons [38]. Some could argue that soccer is a sport practiced on an open field; therefore, vitamin D status could be better. However, the soccer players examined in this research held their practice and training sessions in late afternoon and evening (from 5:00 to 9:00 p.m.), when the sun has set during the winter; hence, it is not likely that the nature of the sport could have positively affected their vitamin D status during this period.

### 4.2. Vitamin D and Physical Fitness in Youth Soccer Players

We found association between serum 25(OH)D levels and results in the S20M, 20Y, and RSI, with better results in fitness tests among those players who had higher levels of 25(OH)D Meanwhile, the findings of previous studies that investigated the effects of 25(OH)D levels on sports performance are not consistent. Koundourakis et al., (2014) found a correlation of sprint and jumping test results with 25(OH)D levels in professional Greek soccer players, whereas several other studies did not [20,25]. Additionally, some studies concluded that soccer players with lower 25(OH)D levels have lower results in performance tests [39]. To explain the possible reasons for our findings (better performance in athletes with higher 25(OH)D levels), a short overview of the mechanisms of the influence of vitamin D on the muscle system and energy capacities of athletes is necessary. Specifically, authors of the study are of the opinion that mechanisms of potential influence of vitamin D on studied performances should be observed from two perspectives: (i) acute influence of higher vitamin D levels on performances, and (ii) chronic (long-term) influence of vitamin D levels on (development) of performance as a result of prolonged period of training. Therefore, the physiological background(s) of those two mechanisms will be presented accordingly.

First, it is important to highlight that vitamin D controls the expression of several proteins that are included in calcium signaling and phosphate-dependent metabolic processes, including ATP and creatine phosphate synthesis in the muscle cells [40]. Vitamin D regulates serum calcium concentrations which directly impacts muscle contraction [41]. Vitamin D increases the influx of calcium into the cytoplasm of muscle cells within minutes by activating two kinases, c-Src and PI3K, enabling calcium to bind to the troponin-tropomyosin complex resulting in exposure to active binding sites and allowing muscle contraction [42]. Increased calcium release and increased myosin movement across actin filaments may result in greater contractile muscle strength [43]. These processes could have all contributed to enhanced performance in tests that require a higher level of muscle excitation (i.e., sprint tests, generic CODS, and RSI). More precisely, all tests where 25(OH)D level was significantly correlated with achievement are dependent of the fast production of force [44]. Therefore, the better the muscular capacity to produce force, the better the sprinting, jumping, and generic-CODS performance. Therefore, there is a certain possibility that higher 25(OH) levels have a positive acute ergogenic impact on explosive performances, and consequently the established correlations should be observed taking into account the previously explained involvement of vitamin D in metabolic (energetic) processes in the human body. As a certain support to such a mechanism of (acute) influence we can note several studies that confirmed the short-term effects of vitamin D supplementation on improvement of explosive muscular capacities in athletes. Specifically, supplementing a high dose (5000–6000 IU per day) of vitamin D for 6–8 weeks among athletes and soccer players led to improved 5 and 10 m sprints, and vertical jumps, but also in aerobic capacity [45,46,47]. Additionally, young soccer players displayed improvements in vertical jumps, triple-hop jump, 10 and 30 m sprints, and shuttle run test after a one-time mega dose (200,000 IU) of vitamin D [23].

However (second), the authors of this study are more of the opinion that the mechanisms of vitamin D influence on fitness capacities should be observed from the perspective of prolonged, long-term effects of better vitamin D availability in athletes who have been involved in a regular training process. Specifically, vitamin D has been shown to play an active role in muscle maturation because, thanks to a vitamin D receptor (VDR)-mediated signal, myoblasts can differentiate into myocytes [48]. Activated VDR acts on cyclin-dependent kinases (serine threonine kinases) that actively participate in cell cycle regulation, stimulating muscle cells to proliferate and differentiate [49]. Vitamin D also regulates the expression of insulin-like growth factor-1, which has a well-recognized role in muscle hypertrophy and remodeling [50,51]. Vitamin D increases the size of myosin heavy chain type II-positive myotubes (e.g., fast-twitch muscle fibers), and increases the diameter and width of fast-twitch fibers [52]. This type of muscle fiber is a major determinant of the explosive type of human movement [53]. As a result, anaerobic activities of maximum intensity (e.g., jumping, running, accelerating) largely depend on the size of fast-twitch fibers. It is also important to note that vitamin D is considered to be involved in the production of testosterone as vitamin D metabolizing enzymes and receptors are expressed also in the Leydig cells where testosterone is produced [54]. It is known that the anabolic action of testosterone is stimulated by transcriptional genes regulation and amino acid uptake, leading to the synthesis of skeletal muscle proteins which later increased power and speed in athletes [45,55]. Supportively, several studies recorded that oral supplementation of vitamin D led to increased testosterone levels [45,56]. In our case it means that there is a certain possibility that players with higher 25(OH)D levels could have higher testosterone levels. It could allow them to undergo increased training load; more importantly, to achieve better supercompensation (supercompensation is a post training period during which the trained function/parameter has a higher performance capacity than it did prior to the training period); and subsequently to improve performance throughout prolonged period of time superiorly than their peers with lower 25(OH)D levels. However, since in this study we observed only 25(OH)D levels, we cannot currently draw a clear conclusion about the correlation between 25(OH)D levels and the studied capacities.

### 4.3. Vitamin D and Motor Performance in Youth Soccer Players

Our results indicated no significant correlation of 25(OH)D levels with motor performances (i.e., agility, balance). The most probable reason for the evidenced results should be found in specifics of soccer-specific performances studied herein. In most common words, these capacities are dependent on skill level, and in youth athletes are not strongly correlated to physical capacities. For example, tests of soccer-specific agility and change of direction speed are highly dependent on the ball-handling skills and perceptual abilities [27,29]. The same applies to balance, which is independent of most of the conditioning capacities, and actually depends on accurate control of body position over the surface [57]. As a result, the eventual positive effect of 25(OH)Don performance that largely depends on skill, and less on physical capacities, is logically limited.

One could argue that CMJ is a test of physical capacity (fitness) and, therefore, should be positively correlated with 25(OH)D. However, we must highlight that although the correlation between CMJ and 25(OH)D do not reach statistical significance when controlled for covariates, the Pearson’s coefficient was actually similar to the correlations between other physical fitness tests (sprinting, jumping, and generic-CODS) and vitamin D. Additionally, CMJ is a relatively non-standard test in soccer players, especially if considering the CMJ-testing sequence used in our investigation (the hands remained on the hips during the test’s execution), which probably resulted in a somewhat lower correlation between 25(OH)D level and CMJ simply because some players who were not highly familiar with the testing did not achieve their best results for CMJ.

### 4.4. Limitations and Strengths

The main limitation of this research comes from its cross-sectional design. Therefore, the results are not the final word of the problem as the causality cannot be determined. Second, we observed only 25(OH)D levels, and we observed no molecular data, which limited the possibility of the more accurate interpretation of the possible mechanisms of the influence of vitamin D on studied performances. Also, players were not supplemented with vitamin D and, therefore, we may not speak about the clear influence of vitamin D status on studied capacities. Furthermore, herein players were not separated according to playing positions since they were not specialized yet, meaning they played several positions in the game. Next, in this study we used 25(OH)D as an indicator of vitamin D status. Although it is globally accepted as the best marker of status of the vitamin D, other measures (i.e., free 25OHD, ratio of 24,25-dihydroxyvitamin D) could eventually be better indicators of vitamin D status, and this issue should be overviewed in future studies. Finally, in this investigation we have used definitions of sufficient/insufficient levels of 25(OH)D which is currently accepted in the country and the region where the study was commenced, but we cannot ignore the fact that there is an ongoing debate regarding the definition of vitamin D deficiency [58].

The main strength of this research is the inclusion of a wide range of performance capacities essential for soccer. Therefore, the results of this research are broadening the knowledge of associations of vitamin D status with numerous performance variables. Furthermore, tested players were from the same soccer club. Thus, the possible influence of other factors on performance is diminished. Therefore, we hope thatour results will improve knowledge in this field, and allow further investigations to be specifically focused on those variables/capacities on which certain influences of vitamin D could be expected.

## 5. Conclusions

Of the studied players, 44% had 25(OH)D insufficiency, which agrees with previous reports. Players were tested at the end of the winter season; although this may at least partially explain these results, the figure is still concerning and points to the necessity of further evaluating the reason for such results even in young athletes. However, only a negligible number of studied players had 25(OH)D deficiency (9.6%), which is a much lower prevalence of deficiency than that evidenced in age-matched youth from the same geographical region. Most probably, the very low 25(OH)D levels are a factor limiting successful participation in competitive sports (and, therefore, youth with very low 25(OH)D do not participate in sports), but it should be studied in more detail in the future.

It appears that 25(OH)D plays a greater role in tests where energetic capacity is essential than in tests where performance skill is crucial. Therefore, results from this research support the theory of the influence of 25(OH)D on the energetic capacity of athletes. Authors are of the opinion that this finding does not necessary imply an ergogenic effect of vitamin D on force production but should rather be observed as a long-term positive effect of appropriate vitamin D levels on supercompensation and consequently on positive training effects during a prolonged period of time (i.e., sports career), resulting in superior sprinting, jumping, and generic-CODS capacity.

However, 25(OH)D was not correlated with tests representing motor performance skill. Considering all the present findings and the results of previous studies, we concluded that 25(OH)D plays a supportive and not a crucial role in sports performance. Future studies should investigate the effects of 25(OH)D supplementation on the performance in various tests of physical capacities among young soccer players and in other sports.

## Figures and Tables

**Figure 1 biology-10-00751-f001:**
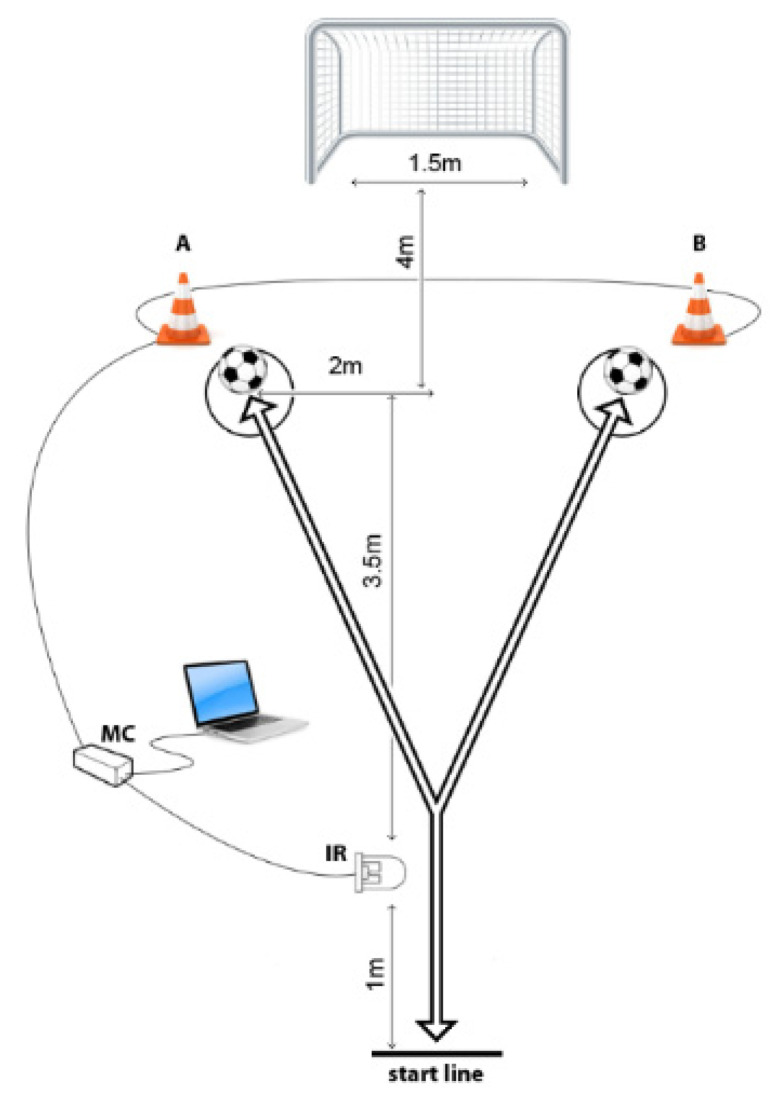
Soccer-specific change of direction speed and agility testing polygon.

**Figure 2 biology-10-00751-f002:**
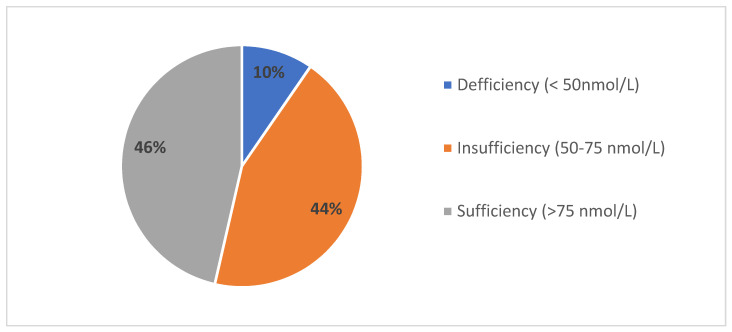
Vitamin D status (25(OH)D levels) in youth soccer players from southern Croatia.

**Figure 3 biology-10-00751-f003:**
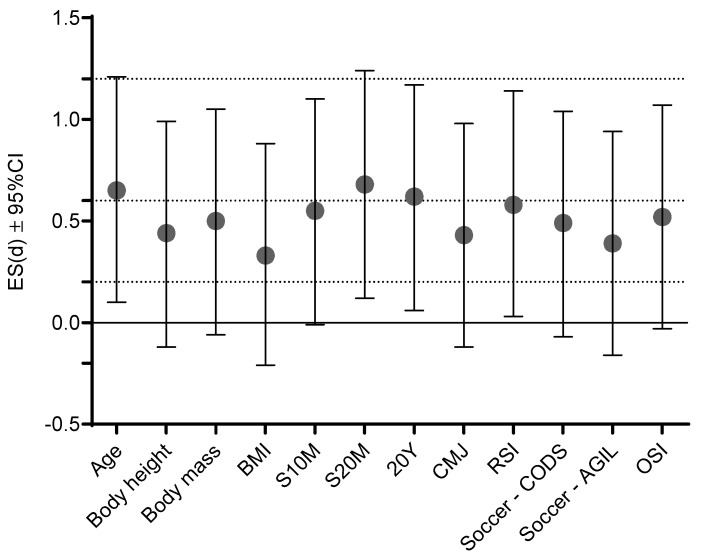
Effect size differences (Cohen’s d) between groups based on vitamin D status (sufficiency vs. insufficiency/deficiency) dashed lines present ES ranges (<0.02 = trivial; 0.2–0.6 = small; >0.6–1.2 = moderate; >1.2–2.0 = large; and >2.0 very large differences).

**Table 1 biology-10-00751-t001:** Descriptive statistics and intra-testing reliability (ICC, CV) for the fitness tests and motor performance tests.

	Mean	Minimum	Maximum	Std.Dev.	ICC	CV
S10M (s)	1.77	1.53	2.06	0.11	0.90	0.03
S20M (s)	3.13	2.72	3.62	0.20	0.94	0.04
20Y (s)	4.86	4.25	5.78	0.36	0.89	0.06
CMJ (cm)	32.23	19.70	44.70	6.04	0.80	0.08
RSI (index)	1.11	0.44	1.79	0.29	0.78	0.08
Soccer-CODS (s)	2.59	2.24	3.29	0.23	0.80	0.08
Soccer-AGIL (s)	2.84	2.39	3.45	0.22	0.77	0.10
OSI (index)	1.63	0.60	6.40	1.03	0.76	0.11

Legend: S10M—sprint 10 m, S20M—sprint 20 m, 20Y—change of direction test over 20 yards, CMJ—countermovement vertical jump, RSI—reactive strength index, Soccer-CODS—soccer specific change of direction, Soccer-AGIL—soccer specific reactive agility test, OSI—overall stability index of balance.

**Table 2 biology-10-00751-t002:** Descriptive statistics and *t*-test differences between groups based on vitamin D status.

	Vitamin D Sufficiency (n = 23)		Vitamin D Deficiency/Insufficiency (n = 29)		*t*-Test	
	Mean	Std.Dev.	Mean	Std.Dev.	*t*-Value	*p*
Age (years)	15.76	1.67	14.37	2.49	2.20	0.03
Body height (cm)	179.38	7.67	182.50	6.64	−1.06	0.30
Body mass (kg)	68.06	9.20	72.50	8.71	−1.22	0.23
BMI (kg/m^2^)	21.09	1.94	21.71	1.77	−0.82	0.42
S10M (s)	1.74	0.11	1.80	0.11	−1.95	0.06
S20M (s)	3.06	0.18	3.19	0.20	−2.45	0.02
20Y (s)	4.74	0.27	4.95	0.40	−2.16	0.04
CMJ (cm)	33.67	6.26	31.09	5.71	1.55	0.13
RSI (index)	1.03	0.28	1.20	0.30	−2.00	0.04
Soccer-CODS (s)	2.52	0.20	2.63	0.25	−1.75	0.09
Soccer-AGIL (s)	2.80	0.15	2.88	0.25	−1.27	0.21
OSI (index)	1.94	1.43	1.39	0.45	1.99	0.05

**Table 3 biology-10-00751-t003:** Pearson’s correlation coefficients between study variables (*** *p* < 0.001, ** *p* < 0.01, * *p* < 0.05).

	1	2	3	4	5	6	7	8	9	10	11	12
Age (1)	-											
Body height (2)	0.03	-										
Body mass (3)	0.38 **	0.74 ***	-									
BMI (4)	0.54 ***	0.23	0.80 ***	-								
25(OH)D (5)	0.33 *	−0.14	−0.14	−0.08	-							
S10M (6)	−0.39 **	0.14	0.08	−0.14	−0.26	-						
S20M (7)	−0.30 *	0.11	0.01	−0.20	−0.39 **	0.94 ***	-					
20Y (8)	−0.47 ***	−0.02	−0.19	−0.20	−0.31*	0.60 ***	0.77 ***	-				
CMJ (9)	0.32 *	−0.06	0.07	0.22	0.27 *	−0.65 ***	−0.73 ***	−0.57 ***	-			
RSI (10)	0.20	−0.15	−0.17	−0.06	0.36 **	−0.45 ***	−0.52 ***	−0.63 ***	0.45 ***	-		
Soccer-CODS (11)	−0.31 *	−0.06	−0.29 *	−0.18	−0.20	−0.59 ***	−0.66 ***	−0.46 ***	0.69 ***	0.41 **	-	
Soccer-AGIL (12)	0.17	0.15	0.25	−0.19	−0.14	0.48 ***	0.62 ***	0.67 ***	−0.52 ***	−0.45 ***	−0.37 **	-
OSI	−0.11	0.30 *	0.44 ***	0.37 **	−0.03	0.34 *	0.49 ***	0.50 ***	−0.32 *	−0.42 **	−0.32**	0.67 ***

## Data Availability

Data will be available upon reasonable request.

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
