# Peer review of "Associations of Vitamin D Levels with Physical Fitness and Motor Performance; A Cross-Sectional Study in Youth Soccer Players from Southern Croatia"

_biology, 2021, doi:10.3390/biology10080751_

Round 1

Reviewer 1 Report

Reviewer’s Comments to Author 

The authors have carried out an investigation on associations of vitamin D levels with physical fitness and motor performance; cross sectional study in youth soccer players 3 from southern Croatia. Overall, the manuscript submitted by Barbara et al. is interesting. The manuscript is organized, and properly referenced. The reviewer thinks that a potential reader can easily understand the details by consulting the main data outlined in the manuscript. The abstract accurately conveys the whole content of the article, and the conclusions are fully appropriate in reviewer’s understanding. The writing is clear and to the point, which is, crucial for a research article. Regarding the writing style is understandable and easy to read for the readership of journal. Tables are also clear and provide a full overview of the trials discussed in the text. For this reason, I consider current version of manuscript adequate to be published in biology journal.  

Minor modifications

Title of the manuscript should include- a cross sectional study rather cross-sectional study……

Line 27, page 1; typo error such as “duet to low” should be removed.

Line 217, Page 5: Why author has used the terminology like -crude values in nmol/L). Either this should be removed or use a term like measured values etc.

Vitamn D levels are key element in this study. How Vitamin D levels were measured? Author did not mention this in manuscript. Which method was used (ELISA based or LC-MSMS based? Proper information regarding the measurement of Vitamin D levels should be included in method and material section. This will benefit the readers.

Author Response

Reviewer #1

Comments and Suggestions for Authors

Reviewer’s Comments to Author 

The authors have carried out an investigation on associations of vitamin D levels with physical fitness and motor performance; cross sectional study in youth soccer players 3 from southern Croatia. Overall, the manuscript submitted by Barbara et al. is interesting. The manuscript is organized, and properly referenced. The reviewer thinks that a potential reader can easily understand the details by consulting the main data outlined in the manuscript. The abstract accurately conveys the whole content of the article, and the conclusions are fully appropriate in reviewer’s understanding. The writing is clear and to the point, which is, crucial for a research article. Regarding the writing style is understandable and easy to read for the readership of journal. Tables are also clear and provide a full overview of the trials discussed in the text. For this reason, I consider current version of manuscript adequate to be published in biology journal.  

RESPONSE: Thank you for recognizing the importance of our research and providing the opportunity to improve the manuscript. We tried to amend the manuscript according to your valuable comments and suggestions.

Minor modifications

Title of the manuscript should include- a cross sectional study rather cross-sectional study……

RESPONSE: Thank you for this comment. Title now reads: Associations of vitamin D levels with physical fitness and motor performance; a cross sectional study in youth soccer players from southern Croatia.

Line 27, page 1; typo error such as “duet to low” should be removed.

RESPONSE: Thank you for this comment, text now reads: “Participants were tested on physical-fitness, motor-performance and vitamin D at the end of winter period, when levels of vitamin D are known to be lowest due to low exposure to sunlight”.

Line 217, Page 5: Why author has used the terminology like -crude values in nmol/L). Either this should be removed or use a term like measured values etc.

RESPONSE: Thank you for this suggestion, we replaced term crude with measured. Text now reads: “To identify the associations between 25(OH)D levels (crude measured values in nmol/L), and physical fitness and motor performance…”

Vitamn D levels are key element in this study. How Vitamin D levels were measured? Author did not mention this in manuscript. Which method was used (ELISA based or LC-MSMS based? Proper information regarding the measurement of Vitamin D levels should be included in method and material section. This will benefit the readers.

RESPONSE: Thank you for this suggestion. The measurement of 25(OH)D is now described more in details including the method used (e.g. Electro-chemiluminescence binding assay - ECLIA). Text now reads: “The 25(OH)D levels were measured using the Elecsys Vitamin D total assay ((Electro-chemiluminescence binding assay – ECLIA), and with a Cobas e601 analyzer (Roche Diagnostics International Ltd., Rotkreuz, Switzerland), using a competitive electrochemiluminescence binding techniqueVitamin D total assay employs a vitamin D-binding protein as the capture protein to bind vitamin D3(25-OH) and vitamin D2(25-OH). The detection range of the test is 7.5–175 nmol/L 25(OH)D, and the sensitivity of the assay is 5 nmol 25(OH)D/L. The intraclass CV ranges from 2.2% (at 174 nmol 25(OH)D/L), to 6.7% (at 165 nmol 25(OH)D/L), with the 5.0 nmol/L, 7.5 nmol/L and 12.5 nmol/L for Limit of Blank, Limit of Detection, and Limit of Quantification, respectively.”  (Please see sections Variables and testing)

Staying at your disposal

Reviewer 2 Report

This is an interesting and well-written study investigating the association of vitamin D status with physical fitness and motor performance in 52 youth soccer players. The authors revealed that serum 25(OH)D level was correlated with sprint 20 meters, 20 yards tests, and RSI, showing a greater role of vitamin D in physical fitness tests. I have a few comments.

  • Since the authors performed comparisons for 8 different outcomes, adjustment for multiple comparisons may be warranted to ensure the validity of statistical analysis.
  • It would be interesting to acquire data on the position of each soccer player, if available, as it may be a conceivable determinant of physical performance which may serve as a confounder or effect modifier for the association.
  • If available, it would also be important to determine estimated vitamin D and calcium intake based on food frequency questionnaire to determine if it correlates with serum 25(OH)D and outcomes.
  • The authors may want to perform multivariate analyses with age, BMI and possibly other baseline characteristics as covariates to determine the effects of confounders.
  • Vitamin D levels should be replaced with 25-hydroxyvitamin D levels as appropriate to avoid confusion.
  • It is unclear if the assay used in this study recognizes both 25(OH)D3 and 25(OH)D2 or there is any selectivity. Please add description also with the % variability of the assay.
  • Please provide the latitude of the study location and comment on how much vitamin D is expected to be produced in the skin during the study period.

Author Response

Comments and Suggestions for Authors

This is an interesting and well-written study investigating the association of vitamin D status with physical fitness and motor performance in 52 youth soccer players. The authors revealed that serum 25(OH)D level was correlated with sprint 20 meters, 20 yards tests, and RSI, showing a greater role of vitamin D in physical fitness tests. I have a few comments.

RESPONSE: Thank you for your valuable comments and support. Also, thank you for recognizing the importance of our research. We tried to amend the manuscript according to your suggestions and hope that we adequately improved it.

Since the authors performed comparisons for 8 different outcomes, adjustment for multiple comparisons may be warranted to ensure the validity of statistical analysis.

RESPONSE: Indeed, the adjustments (such as Bonferoni, for example) could be beneficial, but from our opinion in our study the usage of such statistics will be too “rigorous”, but also rediundant to some extent. Namely, we evaluated different performance variables, which are low correlated (actually, this was one of the main intentions of the study – to observe various capacities, and not those from the same spectre as previous studies regularly did). Also, in this study we evaluated the possible “influence” of vitamin D status by two different approaches: (i) analysis of the differences between groups (done by t-test), and (ii) correlation analysis (Pearson’s correlation). Such “parallel” statistical analyses actually served as “statistical control” of the findings. However, if you will insist on adjustement, we will certainly follow your suggestion; please let us know.

It would be interesting to acquire data on the position of each soccer player, if available, as it may be a conceivable determinant of physical performance which may serve as a confounder or effect modifier for the association.

RESPONSE: Thank you for this suggestion, we did think about it. However, we included young players who are not yet specialized in one playing position and play several positions in the game. Therefore, we did not include the playing position since only few players actually had their main position. It was included in the limitations of the study. Text now reads: “Furthermore, players were not separated according to playing positions since they were not specialized yet, meaning they played several positions in the game”. (Please see section Limitations and Strengths)

If available, it would also be important to determine estimated vitamin D and calcium intake based on food frequency questionnaire to determine if it correlates with serum 25(OH)D and outcomes.

RESPONSE: Thank you for this suggestion. However, we did not give players food frequency questionnaire and we do not have that data. We will definitely include it in future investigations as we consider this very important and interesting.

The authors may want to perform multivariate analyses with age, BMI and possibly other baseline characteristics as covariates to determine the effects of confounders.

RESPONSE. Thank you for your suggestion. In the revised version of the manuscript we paid attention on it, and did amendments accordingly. Therefore, the statistics are done in two phases as stated in the Statistics subsection: “These analyses were done in two phases. In the first phase we calculated simple uni-variate correlations and regression coefficients between all pairs of variables. In the second phase, the  significant correlations (regression coefficients) between 25(OH)D and physical fitness/motor performances that were evidenced in the first phase, were additionally statistically controlled. For this purpose we included age and BMI as co-variates in the correlation analysis, and statistically controlled the confounding effect of age and BMI in calculated correlations between vitamin D and physical fit-ness/motor performance variables. This was done since analyses showed significant correlation between age and vitamin D levels in the first phase, while there was a pos-sibility of the confounding effects on association between 25(OH)D and fit-ness/performance.”

The results are later presented as follows: “Since vitamin D status was significantly (p < 0.05) correlated with participants’ age (Pearson’s R = 0.33, low positive correlation), the significant correlations between fit-ness variables (S20M, 20Y, CMJ, and RSI), and vitamin D status were additionally con-trolled for age as a covariate. Finally, significant (p < 0.05) partial correlations were confirmed for associations between vitamin D and (i) S20M (Pearson’s R = -0.30, low negative correlation), (ii) 20Y (Pearson’s R = -0.31, low negative correlation), and (iii) RSI (Pearson’s R = 0.32, low positive correlation).”

We must note that there was a possibility to perform a multiple regression analysis, but this was limited by number of participants vs. number of variables. Because of that we decided to use the previously specified statistical approach. Thank you!

Vitamin D levels should be replaced with 25-hydroxyvitamin D levels as appropriate to avoid confusion.

RESPONSE: Thank you for this suggestion. We replaced vitamin D levels with 25-hydroxyvitamin D (25(OH)D) levels throughout the manuscript.

It is unclear if the assay used in this study recognizes both 25(OH)D3 and 25(OH)D2 or there is any selectivity. Please add description also with the % variability of the assay.

RESPONSE: Thank you for this suggestion. We added a detailed description of the analysis and assay. Text now reads: The 25(OH)D levels were measured using the Elecsys Vitamin D total assay (Electro-chemiluminescence binding assay – ECLIA) and with a Cobas e601 analyzer (Roche Diagnostics International Ltd., Rotkreuz, Switzerland), using a competitive electrochemiluminescence binding technique. Vitamin D total assay employs a vitamin D-binding protein as the capture protein to bind vitamin D3(25-OH) and vitamin D2(25-OH). The detection range of the test is 7.5–175 nmol/L 25(OH)D, and the sensitivity of the assay is 5 nmol 25(OH)D/L. The intraclass CV ranges from 2.2% (at 174 nmol 25(OH)D/L), to 6.7% (at 165 nmol 25(OH)D/L), with the 5.0 nmol/L, 7.5 nmol/L and 12.5 nmol/L for Limit of Blank, Limit of Detection, and Limit of Quantification, respectively.  (Please see sections Variables and testing)

Please provide the latitude of the study location and comment on how much vitamin D is expected to be produced in the skin during the study period.

RESPONSE: Indeed, we missed to report the exact latitude in the original version. It is now specified, and text reads: “The 52 participants in this cross-sectional research were youth soccer players (15.98 ± 2.26 years old). All players were members of the same soccer team located in Split, Croatia, residing at the latitude of 43° N. “ (please see beginning of the Participants subsection). With regard to “expected values”, to the best of our knowledge there is no clear consensus. However, after we overviewed the studies on children and adolescents done on similar latitudes, it mean values in countries on similar latitudes as Croatia (e.g. Spain and Italy) vary, with 38±13 nmol/L in Italian, and 52±14 nmol/L in Spanish children (https://www.nature.com/articles/s41430-021-00985-4.pdf ). The values are now included in the discussion, and text reads: “The mean value of serum 25(OH)D level of the studied players was 79.03 ± 25.32 nmol/L, with 54% of participants having low 25(OH)D levels (<75 nmol/L). These results are somewhat better than those with previously reported data in Croatia, neighboring countries, and countries with similar latitudes for samples of children and adolescents. For example, very recent report evidenced 38 ± 13 nmol/L in Italian, and 52±14 nmol/L in Spanish children (42° N and 40° N, for Italian and Spanish samples, respectively) (Wolters et al., 2021). Further,  72.3% Bosnian and Herzegovinian (B&H) adolescents (<18 years old) had low 25(OH)D levels (<75 nmol/L) (Sokolovic et al., 2017), whereas 82.2% of Italian adolescents, etc.” (please see 1st paragraph of the subsection 4.1 Vitamin D status in youth soccer players).

 Staying at your disposal.

Reviewer 3 Report

Major comments:

  1. VDR expression in muscle is not as high as claimed, please refer to data of the human protein atlas.
  2. The main problem of this manuscript is that it provides only a low amount of data. None of them is molecular and there is again of mechanistic insight.
  3. The definition of a vitamin D status of 50-75 nM as insufficient is questionable. If you follow the Institute of Medicine's definitions, this applies to a range of 25-50 nM and already levels above 50 nM are sufficient. This would change the readout of the whole manuscript.

Minor comments:

  1. Abbreviations should be defined at first use (e.g. 25(OH)D in the Abstract) and used then consistently throughout the manuscript.
  2. In case vitamin D3 or its metabolites are meant, please indicate this by a respective index. Is vitamin D2 also measured?
  3. There is only one vitamin D receptor, no plural.

Author Response

Comments and Suggestions for Authors

Major comments:

  1. VDR expression in muscle is not as high as claimed, please refer to data of the human protein atlas.
  2. The main problem of this manuscript is that it provides only a low amount of data. None of them is molecular and there is again of mechanistic insight.
  3. The definition of a vitamin D status of 50-75 nM as insufficient is questionable. If you follow the Institute of Medicine's definitions, this applies to a range of 25-50 nM and already levels above 50 nM are sufficient. This would change the readout of the whole manuscript.

RESPONSE: Thank you for your comments. Indeed, we are aware that there are several different definitions of vitamin D status, and “measurements”. However, we used the approach and references of Society for Adolescent Health and Medicine, and Endocrine Society that consider 25(OH)D concentrations <75nmol as insufficient:

  • Society for Adolescent Health and Medicine. Recommended vitamin D intake and management of low vitamin D status in adolescents: a position statement of the society for adolescent health and medicine. J Adolesc Health. 2013 Jun;52(6):801-3. doi: 10.1016/j.jadohealth.2013.03.022.
  • Holick MF, Binkley NC, Bischoff-Ferrari HA, Gordon CM, Hanley DA, Heaney RP, Murad MH, Weaver CM; Endocrine Society. Evaluation, treatment, and prevention of vitamin D deficiency: an Endocrine Society clinical practice guideline. J Clin Endocrinol Metab. 2011 Jul;96(7):1911-30. doi: 10.1210/jc.2011-0385.

What is particularly important for our study, we must note that the previously specified guidelines are adopted in Croatian recommended levels of 25(OH)D, and are therefore regularly used in studies in our country (and our region), including studies where athletes were specifically observed, for example:

  • https://pubmed.ncbi.nlm.nih.gov/32783677/
  • https://pubmed.ncbi.nlm.nih.gov/32724281/
  • https://pubmed.ncbi.nlm.nih.gov/31766877/
  • https://www.ncbi.nlm.nih.gov/pmc/articles/PMC6780345/

Therefore, we hope that you will understand our current position, and will agree that eventual questioning of the (officially) accepted norms could be as (at least) problematic, especially considering the fact that some of the authors are clinicians.

However, we absolutely agree that there are strong evidences the current norms should be eventually re-evaluated, and consequently in this version of the paper we included specific part in the limitations subsection where the problem of measurement and definition of vitamin D status is accentuated. Text reads: “Next, in this study we used 25(OH)D as an indicator of vitamin D status, which is globally accepted as the best marker of status of the vitamin D. However, other measures (i.e. free 25OHD, ratio of 24,25-dihydroxyvitamin D [24,25(OH)2D]:25OHD) could eventually be better indicators of vitamin D status, which should be overviewed in future studies. Finally, we can not ignore the fact that there is an ongoing debate regarding the definition of vitamin D deficiency (Giustina et al., 2020). In this investigation we have used definitions which is currently accepted in the country and the region where the study was commenced. In further investigations, other definitions should be used and evaluated as well. “ (please see Limitations section).

Also, we tried to highlight that the current controversies about vitamin D measurement and levels probably didn’t greatly influence our investigation since (cited) “… , in this study we: (i) compared 25(OH)D levels with previous reports which used same measurement protocols and standards, and (ii) evaluated the associations between 25(OH)D levels and fitness/performance in youth soccer players.” (please see Limitations section).

Minor comments:

  1. Abbreviations should be defined at first use (e.g. 25(OH)D in the Abstract) and used then consistently throughout the manuscript.

RESPONSE: Thank you for this suggestion, we tried to follow it throughout the manuscript.

  1. In case vitamin D3 or its metabolites are meant, please indicate this by a respective index. Is vitamin D2 also measured?

RESPONSE: Thank you for this observation. Indeed, in the original version we missed to specify that total assay employs a vitamin D-binding protein as the capture protein to bind vitamin D3(25-OH) and vitamin D2(25-OH).  Therefore, both D2 and D3 were measured. Text now reads: “The 25(OH)D levels were measured using the Elecsys Vitamin D total assay (Electro-chemiluminescence binding assay – ECLIA), and with a Cobas e601 analyzer (Roche Diagnostics International Ltd., Rotkreuz, Switzerland), using a competitive electrochemiluminescence binding technique. Vitamin D total assay employs a vitamin D-binding protein as the capture protein to bind vitamin D3(25-OH) and vitamin D2(25-OH). “ (please see 3rd paragraph of the Variables subsection)

  1. There is only one vitamin D receptor, no plural.

RESPONSE: Thank you for this comment, we corrected it in the text. Actually, in the original version the plural was a result of “grammatical check” (i.e. “large number of vitamin D receptor(s)”).

Staying at your disposal.

Round 2

Reviewer 3 Report

The authors missed to address my most important major point 2.

Author Response

COMMENT: The authors missed to address my most important major point 2.

RESPONSE: First of all, please accept our apology for not addressing your 2nd comment in the 1st revision (e.g. “The main problem of this manuscript is that it provides only a low amount of data. None of them is molecular and there is again of mechanistic insight”) - it was absolutely unintentionally. Bein honest, the first review was comprehensive, so we were actually "lost" in changes. In this version we tried to overview the specified problem more accurately and specifically.

  1. With regard to your statement of “low amount of data”; we couldn’t agree more with regard to “biochemical” results. However, we tried to point that our main intention was to evaluate the (specific) associations between 25(OH)D and various (numerous) conditioning capacities, which was not the case in previous studies where authors were oriented toward some specific conditioning capacities (almost exclusively fitness-, and not performance/skill-related capacities). Also, if we may accentuate, this is one of the rare studies where players from (only) one team were observed (previous studies generally observed participants from various teams), meaning that the potential influence of training methodology is limited. However, the fact that we observed only limited amount of (non molecular) data is now specified in the limitations subsection (please see highlighted text)
  2. With regard to “mechanisms” of possible influence of vitamin D on studied performances, in this version we made substantial changes in the manuscript. Specifically, the whole subsection 4.2. is now devoted to potential mechanisms of influence of vitamin D levels on studied fitness capacities (when it comes to those where we evidenced certain correlations between vitamin D and achievement). In brief, we tried to overview two possible mechanisms which could be the base of the established associations: (i) the acute influence of higher vitamin D levels on fitness, and (ii) chronic (prolonged) influence of higher vitamin D levels on efficacy of training (which consequently allowed better long-term development). Please excuse us for not copying the text here (it is relatively long – 1.5 pages), but please see the text highlighted in yellow in Discussion section. Also, the background of mechanics of possible background is briefly presented in conclusion section (highlighted text).

Staying at your disposal, and thank you once again.

Authors

Authors

Round 3

Reviewer 3 Report

no further comments